Crop damage by vertebrates in Latin America: current knowledge and potential future management directions

Cuesta Hermira Adrián Alejandro 1 2 3
http://orcid.org/0000-0002-8074-9964 Michalski Fernanda 2 4 5 fmichalski@unifap.br
1 Ecology Department, Federal University of Rio Grande do Sul , Porto Alegre, RS , Brazil
2 Ecology and Conservation of Amazonian Vertebrates Research Group, Federal University of Amapá , Macapá, Amapá , Brazil
3 Centre for Functional Ecology, Department of Life Sciences, University of Coimbra , Coimbra , Portugal
4 Postgraduate Programme in Tropical Biodiversity, Federal University of Amapá , Macapá, Amapá , Brazil
5 Pro-Carnivores Institute , Atibaia, São Paulo , Brazil
Sosa Victoria
Electronic publication date: 2022 Mar 25
Publication date: 2022
Volume: 10
Electronic Location ID: e13185
Received 2021 Oct 18; Accepted 2022 Mar 8
Copyright: © 2022 Alejandro Cuesta Hermira and Michalski
Copyright year: 2022
Copyright holder: Alejandro Cuesta Hermira and Michalski
License: This is an open access article distributed under the terms of the Creative Commons Attribution License, which permits unrestricted use, distribution, reproduction and adaptation in any medium and for any purpose provided that it is properly attributed. For attribution, the original author(s), title, publication source (PeerJ) and either DOI or URL of the article must be cited.
License URL: https://creativecommons.org/licenses/by/4.0/

Keywords: Human-wildlife conflict, Agri-environment schemes, Animal damage, Bird damage, Mammal damage, Crop feeding, Crop protection

Funding: Fundación Mutua Madrileña (Beca de Posgrado 2020) National Council for Scientific and Technological Development (CNPq) 302806/2018-0 University of Coimbra The data was collected during Adrián Alejandro Cuesta Hermira’s masters study, which was funded by a studentship from the Fundación Mutua Madrileña (Beca de Posgrado 2020). Fernanda Michalski receives a productivity scholarship from the National Council for Scientific and Technological Development (CNPq—process 302806/2018-0). This work was also funded by the University of Coimbra via the "International Master in Applied Ecology". The funders had no role in study design, data collection and analysis, decision to publish, or preparation of the manuscript.

==============================
Background

Crop farming contributes to one of the most extensive land use activities in the world, and cropland areas continue to rise. Many vertebrate species feed on crops, which has caused an increase in human-wildlife conflicts in croplands. Crop-feeding damages the economy of local communities and causes retaliation against the responsible vertebrates in several forms, including lethal practices such as hunting and poisoning. Lethal control may cause the local extirpation of some species, affecting ecological processes and patterns. Therefore, it is necessary to find non-lethal alternatives that can protect both local economies and wildlife. Research has been conducted in Africa and Asia, focusing on elephants and primates, and the effectiveness of some non-lethal alternatives, such as chili-based repellents and beehives, is being investigated. However, there has been very little research on this topic in Central and South America. The goal of this review is to assess the current knowledge on crop damage by vertebrates in Central and South America and indicate future research directions.

Survey methodology

We reviewed the available scientific literature reporting crop damage by vertebrates in Central and South America, and the Caribbean, published between 1980 and 2020, through systematic searches on Web of Science, Scopus, and Google Scholar. We analyzed the temporal and geographical distributions of the studies, the crops and vertebrate species these studies considered, the crop protection techniques used, and their effectiveness.

Results

We retrieved only 113 studies on crop damage by vertebrates in Latin America, but there was an increasing trend in the number of studies published over time. Most of the studies were conducted in Brazil, Argentina, Mexico, and Costa Rica. Four orders of mammals (Rodentia, Carnivora, Artiodactyla, and Primates) and four orders of birds (Passeriformes, Columbiformes, Psittaciformes, and Anseriformes) were the most common groups of crop-feeding vertebrates. The most prominent crop was corn, which was featured in 49% of the studies. Other notable crops include rice, sorghum, and sugarcane. The most reported method for protecting crops was lethal control through hunting or poisoning. Non-lethal techniques were found to be less prevalent. Less than half of the studies that mentioned the use of protection techniques indicated their effectiveness, and only 10 studies evaluated it by performing scientific experiments and reporting their results.

Conclusions

Central and South America is still underrepresented in research on vertebrate crop-feeding. There is a need for experimentation-based robust research to find crop protection techniques that minimize harm to vertebrates while effectively reducing damage to crops. While this is being studied, habitat loss and fragmentation need to be halted to prevent the native vertebrates from turning to crops for food.

Introduction

Agriculture accounts for one of the most extensive uses of land; in 2015, it covered ∼37.4% of the global land area, of which 12.2% was dedicated to crops, occupying an estimated 1.6 billion hectares worldwide, of which 198 million hectares (Mha) are in Latin America (Goldewijk et al., 2017). A recent study showed that between 2000 and 2019, the global cropland surface increased by 9% (corresponding to an addition of over 100 Mha), and South American cropland increased by 50% during the study period (Popkin, 2022). The amount of land needed for crops in the future depends largely on how global societies and economies develop (Stehfest et al., 2019). O’Neill et al. (2017) proposed five diverging narratives for global development in the 21st century, called the shared socioeconomic pathways. If global development continues without drastic changes, the land surface for growing crops will need to increase by up to 400 Mha by 2100, with the worst-case scenario requiring an increase of up to or above 700 Mha (Riahi et al., 2017). The projected change in cropland cover is not homogeneous worldwide, and it depends on the role of different regions. Latin America is a perfect example of a “producer” region where cropland cover and production are expanding largely for exportation to “consumer” regions (Europe and North America), where agricultural land cover is decreasing (Creutzig et al., 2019).

One of the reasons for the requirement of such massive land covers is the inefficiency of the production system; much of the product is either lost or wasted. It has been estimated that 73% of the net primary production of global croplands is lost before harvest (Alexander et al., 2017). Damage by living organisms is one of the leading causes of crop losses worldwide; pathogens and pests are estimated to cause global yield losses of 17.2% to 30% in five major crops (wheat, rice, maize, potato, and soy) (Savary et al., 2019). There is a lack of information about the global crop losses caused by vertebrates, but damage to crops caused by birds and mammals is one of the most common factors of conflict between humans and vertebrates worldwide (Torres, Oliveira & Alves, 2018). Climate change is anticipated to cause increased crop losses in the future by increasing the incidence of pests (Deutsch et al., 2018); increase the frequency and intensity of extreme weather events that reduce crop production (Lesk, Rowhani & Ramankutty, 2016); and drive a change to less efficient cultivation practices (Tito, Vasconcelos & Feeley, 2018).

The expansion of human activities and the intensification of land use encroachment on pristine areas via alteration of their extent and distribution causes habitat loss and fragmentation, which may change the distribution and abundance of vertebrate species (Ramesh & Downs, 2015; Said et al., 2016; Zhang et al., 2017). The reduction in food sources due to habitat loss and degradation encourages wild animals to feed on crops, increasing their interactions with human communities, and human-wildlife conflicts (Jorgenson & Sandoval, 2005; McKinney, 2019; Mekonnen et al., 2018). Crop-feeding compromises the food security of local communities and damages economies relying on agriculture (Barirega et al., 2010; Gontse, Mbaiwa & Thakadu, 2018; Hill, 2000; Raphela & Pillay, 2021). Additionally, it represents a serious problem for conservation efforts by reducing human tolerance to wildlife (Campbell-Smith et al., 2010; Sifuna, 2005; Virtanen et al., 2021), turning farmers against conservation initiatives (Dakwa, 2016; Mogomotsi et al., 2020; Osborn & Parker, 2003; Redpath, Bhatia & Young, 2015), and putting crop-feeding species in danger of retaliatory actions by farmers (Compaore et al., 2020; Kendall, 2011; Zimmermann et al., 2009).

When suffering from crop damage, farmers may favor lethal action against the culprit species to prevent further economic losses (Abrahams, Peres & Costa, 2018; Canavelli, Swisher & Branch, 2013; Cossios, Ridoutt & Donoso, 2018; Lima et al., 2019; Linz et al., 2015) or to make a compensatory profit (Scotson, Vannachomchan & Sharp, 2014). If a species is vulnerable, such as rare or slowly reproducing species, retaliatory culling may result in local extirpation of the species (Hockings & McLennan, 2016). Such extinctions may have far-reaching effects on the ecosystem if the species is an essential part of the food web or plays important ecological roles, such as seed dispersal, with their disappearance causing cascading effects on the community (Castillo-López et al., 2017). Furthermore, the use of poison to kill crop-damaging vertebrates can have severe consequences not only for the target species (Lima et al., 2019), but also for other animals that may consume them, such as predators or scavengers (Baudrot et al., 2020; Kalaivanan et al., 2011), and affect the health of human communities and cause social conflicts (Rani et al., 2021). Thus, there is a need to find alternative, non-lethal crop protection techniques that can effectively protect crops while preserving the consumer vertebrate species that damage them. By mitigating crop-feeding conflicts, local economies can be protected, while reducing the risks for wildlife conservation (King et al., 2017).

Studies on human-wildlife conflicts involving crops have been mostly concentrated on two vertebrate groups that have long been considered agricultural pests and have caused global concern: rodents (Capizzi, Bertolino & Mortelliti, 2014; Lauret et al., 2020; Stenseth et al., 2003) and birds (Anderson et al., 2013; de Mey, Demont & Diagne, 2012; Kale et al., 2014; Montràs-Janer et al., 2019). Crop protection techniques against birds and rodents have traditionally involved lethal control through population suppression (Capizzi, Bertolino & Mortelliti, 2014; Linz et al., 2015), even though research on the alternative non-lethal techniques, such as the use of chemical repellents has been conducted (DeLiberto & Werner, 2016). Other vertebrate groups, such as elephants (Mayberry, Hovorka & Evans, 2017; Naughton-Treves & Treves, 2005; Nsonsi et al., 2018; Sitati et al., 2003) and more recently primates (Hockings & Sousa, 2013; Marchal & Hill, 2009; Mc Guinness & Taylor, 2014; Priston, Wyper & Lee, 2012; Siljander et al., 2020; Wallace & Hill, 2012) are at the center of increased concern regarding crop-feeding in Africa and Asia (Siljander et al., 2020). Non-lethal protection techniques that are effective in deterring crop-feeding by elephants include chili (Capsicum) based methods (Chang’a et al., 2016; Osborn, 2002), use of beehives (King, Douglas-Hamilton & Vollrath, 2011; King et al., 2009; Ngama et al., 2016), and playing of predator growls (Thuppil & Coss, 2016). There are examples of chili-fences failing to increase the proportion of elephant attacks repelled (Gunaryadi & Hedges, 2017; Hedges & Gunaryadi, 2010) and of beehives not preventing occasional widespread damage to crops (Kiffner et al., 2021), but overall these techniques have been proven to help protect local livelihoods and conserve wildlife (Chang’a et al., 2016; King et al., 2017). Despite garnering significant attention in recent years, very few non-lethal protection techniques against primates have been tested, with some exceptions such as the use of nets, which are effective in reducing fruit consumption by orangutans (Campbell-Smith, Sembiring & Linkie, 2012), and preliminary trials have been conducted on using plant substances as feeding deterrents for macaques (O’Brien & Hill, 2018).

The scientific literature on crop-feeding is less abundant in Latin America than in Africa and Asia. From the available literature, only a few studies reference crop protection techniques, most of which focus on lethal methods such as hunting (Cossios, Ridoutt & Donoso, 2018; Naughton-Treves et al., 2003; Rosa, Wallau & Pedrosa, 2018) or poisoning (Espinoza & Rowe, 1979; Villafaña Martín et al., 1999). The use of lethal controls to manage the populations of crop-damaging birds on the continent has been shown to be ineffective (Linz et al., 2015). Moreover, the development and testing of non-lethal crop protection techniques that could be effective in Latin America are lacking. However, there are a few studies that have indeed tested the effectiveness of non-lethal crop protection techniques through scientific experiments (Avery, Tillman & Laukert, 2001; Castillo-López et al., 2017; Mitchell & Bruggers, 1985; Pérez & Pacheco, 2006, 2014; Robles et al., 2003; Rodriguez et al., 1995).

In this study, we review the published literature on crop damage by vertebrates in Latin America. The rationale of this study stems from the need to collect the available scientific knowledge on the topic, set the groundwork for future research that could lead to the development of effective non-lethal protection techniques, and mitigate human-vertebrate conflicts in Latin America. We attempted to determine which groups of vertebrates are most involved with crop-feeding, assess the effectiveness of different crop protection techniques, and highlight key knowledge gaps. This review could be useful to a broad audience, from researchers and conservation practitioners, to subsistence and commercial farmers.

Survey Methodology

We reviewed the available scientific literature reporting crop damage by vertebrates in Central and South America, and the Caribbean. We followed the Preferred Reporting Items for Systematic Reviews (PRISMA) guidelines (Page et al., 2021). One of the authors (ACH) conducted systematic searches of three databases: Scopus, Web of Science (core collection), and Google Scholar in November 2021. In Scopus and Web of Science search strings were created using three categories of terms (vertebrates, crop damage, and location) with Boolean operators AND between categories and OR within categories: “Vertebrate*” or “Wildlife” or “Mammal*” or “Bird*” or “Reptile*” or “Amphibian*” or “Fish*”, “Crop*” or “Crop damage*” or “Crop raid*” or “Crop loss*” or “Crop protection” or “Agriculture” or “Subsistence”, and “Neotropic*” or “South America” or “Central America” or “Mexico” or “Guatemala” or “Honduras” or “Panama” or “Caribbean” or “Nicaragua” or “El Salvador” or “Costa Rica” or “Venezuela” or “Colombia” or “Ecuador” or “Guyana” or “French Guiana” or “Suriname” or “Brazil” or “Peru” or “Bolivia” or “Chile” or “Argentina” or “Paraguay” or “Uruguay.” This search string was applied to study titles, abstracts, and keywords. In Google Scholar, a total of 1,176 possible combinations of terms from the three categories were searched individually, and Publish or Perish software (Harzing, 2007) was used to retrieve the search results. The searches on the three databases covered a publishing period of four decades (1980–2020). Searches were performed only in English, but when studies written in Spanish, Portuguese, or French were returned, they were also considered for the review.

The titles and abstracts of all results returned by the searches were screened for relevance. Only records of studies that were performed in Central America, South America, and the Caribbean, which fully or partially focused on crop-damaging vertebrate species, and/or crop protection techniques used against them were retained. Records that did not meet these criteria were excluded. A similar procedure was used for the results returned from Google Scholar; however, only the first 50 records obtained from each search were considered. Systematic reviews commonly conduct searches solely on commercial databases (e.g., Scopus and Web of Science) (Haas & Lortie, 2020; Miguel, Butterfield & Lortie, 2020; van Wilgen et al., 2018). We chose to use Google Scholar also as it forms a powerful addition to other traditional search methods (Haddaway et al., 2015). While searching for records in Google Scholar, systematic reviews typically screen the first 50–100 search records (Duarte, Norris & Michalski, 2018; Haddaway et al., 2015; Hughes et al., 2014). The authors (ACH and FM) conducted independent reviews of the studies assessed for eligibility during the screening phase and discarded Ph.D. or MSc theses, technical reports, and off-topic studies. Although gray literature can have relevant data and information, we found that adding it to systematic reviews has drawbacks. The main challenge is associated with limited time and resources (Mahood, Van Eerd & Irvin, 2014) as searches in multiple search engines may be required (Paez, 2017). Additionally, adding gray literature to systematic reviews may introduce problems related to the reproducibility of methodologies to be systematic, as there is scant information about how searches for gray literature are executed (Mahood, Van Eerd & Irvin, 2014). However, in order to minimize bias in our systematic review, we included conference proceedings (McAuley et al., 2000). The number of studies excluded and retained was recorded for each of the screening stage according to the PRISMA statement (Page et al., 2021).

The selected studies were sorted into one or more of the following categories: (1) Crop damage evaluation, if the damage caused to crops by vertebrates in the area was assessed; (2) Crop protection experiment, if an experiment testing the effectiveness of crop protection techniques was performed; (3) Protection technique evaluation, if the study analyzed the effectiveness or feasibility of a particular protection technique but no experiment was conducted; (4) Farmer perception, if interviews with local farmers were used to assess their knowledge and/or opinions; (5) Pest species or outbreak overview, if the article reported the general information about one or several species considered as pests or on specific outbreaks; (6) Crop-feeding species behavior, if the study focused on the diet or other behavioral aspects of the vertebrate species.

ACH extracted the following data from the selected studies: (a) date of publication; (b) country or countries where the study took place; (c) geographical coordinates of the study sites; (d) presence or absence of maps of the study area; (e) type of plantation (commercial, subsistence, or other); (f) crop species included in the study; (g) crop-damaging vertebrate species or taxa included in the study; (h) methods used to identify the vertebrate taxa; (i) methods used to quantify crop damage; (j) methods used to reduce damage to crops; and (k) effectiveness of the protection methods (effective, not effective, or not evaluated). The lack of data or unclear information was also noted. We classified a study under subsistence plantations when this was explicitly mentioned in the article or when it was implied that all or most of the crops produced were used to maintain the farmer’s family and community. Studies were classified as being conducted on commercial plantations when the article implied that the crops were raised mainly for economic profit. The techniques used to reduce crop damage were classified into 13 categories: hunting, poisoning, biological control, reproductive control, chemical repellents, agricultural practices, vigilance, physical barriers, acoustic deterrents, visual deterrents, olfactory deterrents, palatable deterrents, and capture and relocation. The protection techniques evaluated in each study were considered “effective” or “not effective” when the study provided experimental results regarding the effectiveness of the techniques, when the study included interviews with farmers concerning the effectiveness of the techniques, or when the study showed other evidence attesting to the effectiveness of the techniques. Otherwise, the effectiveness of the reported techniques was classified as “undetermined”.

The vertebrate species and taxa were grouped by taxonomical order. Importance values were calculated for each order. The number of taxa in each order featured in each article was counted, and the totals were summed to produce the final importance value assigned to each order. Thus, every appearance of a taxon of the same order in an article was counted. An ecological network figure showing the interactions between vertebrate orders and crop genera was plotted using the bipartite package in R (Dormann, Gruber & Fründ, 2008). For this purpose, when there was more than one vertebrate taxon of the same order in a study, it was considered a single interaction. The importance values attributed to the different vertebrate orders did not always correlate with their weight in the interaction network, as these parameters represent two different traits of the orders. The importance value reflects the number of appearances of each order’s taxa in the reviewed literature, whereas, for the network, only one interaction between a vertebrate order and a crop genus was counted per study, independent of the number of taxa from that order that were reported in the study. Thus, orders that have a high importance value because they are widely represented in the literature but only interact with a few crop genera, will have comparatively little weight on the interaction network. The status category of all vertebrate species that could be identified in this review falls under the IUCN Red List of Threatened Species (IUCN, 2021).

The geographic coordinates of the studies were used to produce a distribution map using ArcGIS 10.5.1 (ESRI, 2017). When studies failed to provide the exact geographic coordinates of the study area, we used Google Earth to obtain the approximate coordinates with the help of the maps of the study area and/or key landmarks, such as towns or protected areas, which could be clearly distinguished on Google Earth images. When studies provided geographical coordinates in another system, they were converted to decimal degrees. For studies with more than one coordinate in the same study area, we plotted the mean position between the study sites (Laufer, Michalski & Peres, 2013). When studies reported more than one study area and the distance between them was greater than 50 km, we plotted more than one point for the same study (Duarte, Norris & Michalski, 2018). The locations of the study sites were plotted over satellite-derived cover and shaded relief data with ocean bottom from the Natural Earth Dataset (http://naturalearthdata.com/) and other freely available data on cropland distribution (Massey et al., 2017).

Results

Compilation of studies

The searches returned 118 records that met all initial selection criteria, and an additional four records previously known to the authors were later included. Of the 122 records, seven were excluded because they were gray literature: six were MSc or Ph.D. theses, and one was a technical report. Two additional records were excluded because they studied damage by vertebrates to silo bags (Zufiaurre, Abba & Bilenca, 2020) and farming machinery (Álamo Iriarte, Sartor & Bernardos, 2019) and not to crops directly. After this process, 113 studies were included in the final analysis (Fig. 1).

Figure 1 PRISMA flow diagram for the systematic review included in the analyses.

Geographic and temporal distribution of studies

The temporal distribution of studies showed that there has been an increase in the number of studies published on the topic since the 1980s. Only 21 studies (19%) were published before 2000 and 64 (57%) were published in the last decade (2011–2020). The year with the most studies published was 2018, with 11 (10%) studies (Fig. 2). The studies were scattered across most Central and South America, with some in the Caribbean (Fig. 3). The study sites were located across areas with different proportions of land cover by crops (Fig. 3). The countries with the highest number of studies were Brazil (n = 31), Argentina (n = 18), and Mexico (n = 13). Other countries with study sites included Costa Rica (n = 10); Peru (n = 9); Venezuela (n = 8); Bolivia, and Uruguay (n = 7); Colombia (n = 6); Barbados (n = 3); Cuba, Chile, and Puerto Rico (n = 2); and Ecuador, Guyana, Belize, Dominican Republic, and Saint Kitts and Nevis (n = 1) (Table S1). Four studies were conducted in more than one country.

Figure 2 Annual number of studies on crop damage by vertebrates in Latin America from 1980 to 2020.

The color gradient is proportional to the number of studies in each year. The blue line depicts the trendline and the shaded area represents the 95% confidence interval.

Figure 3 Spatial distribution of studies on crop damage by vertebrates in Latin America.

The white circles represent the locations of the study sites for each article. The magenta areas represent surface covered by crops.

Type of crop plantations and studies

The majority of the studies were conducted on commercial plantations (n = 60, 53%), followed by subsistence or semi-subsistence plantations (n = 23, 20%). The remaining studies were conducted in experimental fields (n = 6), laboratories (n = 2), harvesting concessions (n = 1), and areas for which the type of plantation could not be reliably determined (n = 22).

Among the six categories in which the studies were categorized based on their focus, the one with the highest number of studies was “Crop damage evaluation” with 42 studies (37%), followed by “Farmer perception” with 38 studies (34%), and “Crop-feeding species behavior” with 37 studies (33%). The other three categories had fewer studies: “Pest species or outbreak overview” included 16 studies (14%), “Crop protection experiment” had 10 studies (9%), and “Protection technique evaluation” included six studies (5%) (Table S1).

Vertebrates and crops

In total, 272 crop-damaging vertebrate taxa were studied in 113 reviewed studies; these included mammals, birds, and reptiles (Table S2). The number of taxa included in each study varied greatly, ranging from 1 to 56, with 64 (57%) studies focusing on a single vertebrate species. The mammal taxa represented nine different orders: Rodentia (70 taxa), Primates (12 taxa), Carnivora (11 taxa), Cingulata (8 taxa), Artiodactyla (7 taxa), Didelphimorphia (6 taxa), Lagomorpha (4 taxa), Perissodactyla (2 taxa), and Chiroptera (1 taxon). The bird taxa represented 13 orders: Passeriformes (77 taxa), Psittaciformes (22 taxa), Columbiformes (18 taxa), Anseriformes (11 taxa), Piciformes (7 taxa), Gruiformes (4 taxa), Galliformes (3 taxa), Accipitriformes (2 taxa), Charadriiformes (2 taxa), Cariamiformes (1 taxon), Cuculiformes (1 taxon), Pelecaniformes (1 taxon), and Strigiformes (1 taxon). Finally, there was only one reptile taxon in the order Squamata (Table S2). The most represented order among the reviewed studies was Rodentia, with an importance value of 126, followed by Passeriformes (123), Columbiformes (55), Psittaciformes (37), Carnivora (30), Artiodactyla (26), Primates (20), and Anseriformes (15) (Fig. 4). Together, these eight orders accounted for almost 90% of the representation in all studies. The remaining orders had importance values below 10 (Fig. 4).

Figure 4 Importance value of the vertebrate orders represented in the studies.

The importance value was calculated by counting the number of taxa of each vertebrate order featured in each article and then summing the totals. Mammal, bird, and reptile orders are represented in purple, yellow, and green, respectively.

Across all reviewed studies, 77 genera of crops were reported to be damaged by vertebrates, with the number of genera per study varying from 1 to 31. Forty-six studies included only one crop genus, and 12 did not specify which crops were affected by vertebrates. The most prominent crop in the studies was corn (Zea sp.). It was featured in 55 (49%) studies and interacted with 17 of the 23 vertebrate orders represented in our review, most predominantly with Rodentia, Psittaciformes, and Artiodactyla (in 18, 16, and 15 studies, respectively) (Fig. 5). The second most represented crop was rice (Oryza sp.), which appeared in 28 (23%) studies and was mainly damaged by Passeriformes and Rodentia (11 studies each) (Fig. 5). Sorghum (Sorgum sp.) reportedly suffered damage in 23 (20%) studies, mainly by three bird orders: Columbiformes (nine studies), Passeriformes and Psittaciformes (six studies each). Sugarcane (Saccharum sp.) was mentioned in 19 (17%) studies and interacted with four mammalian orders: Rodentia (9), Primates (seven studies), and Artiodactyla and Carnivora (two studies each) (Fig. 5). Both, soy (Glycine sp.) and banana (Musa sp.), were mentioned in 15 (13%) studies. Soy was damaged mostly by Columbiformes (seven studies), and Artiodactyla and Psittaciformes (three studies each) (Fig. 5). Bananas interacted the most with Primates (six studies), Carnivora (5), and Rodentia (4) (Fig. 5). Wheat (Triticum sp.), beans (Phaseolus sp.), and sunflowers (Helianthus sp.) were mentioned in 14 (12%) studies. Damage to wheat has been reported mainly by Columbiformes (seven studies). Beans suffered damage mainly by Rodentia (six studies). Sunflowers mostly interacted with Psittaciformes (eight studies) and Columbiformes (5). Finally, manioc (Manihot sp.) was included in 12 (11%) studies and was damaged mainly by Artiodactyla and Rodentia (six studies each) (Fig. 5).

Figure 5 Network of interactions between vertebrate orders and crop genera found within the 113 studies included in this review.

Each article in which a vertebrate order was documented to cause damages to a crop genus is counted as one interaction. The width of the nodes is proportional to the number of interactions that each crop genus or vertebrate order had in total. Similarly, the width of each link is proportional to the number of interactions of its particular pair.

Of the 238 vertebrate taxa that could be identified at the species level from our review, 22 were not categorized as Least Concern (LC), four species considered Endangered (EN), eight species considered Vulnerable (VU), seven species categorized as Near Threatened (NT), and three as Data Deficient (DD) (IUCN, 2021). Of these species, four were birds, three were psittacines (VU – Eupsittula canicularis, NT – Amazona aestiva and Eupsittula nana), and one was from the Galliformes order (NT – Colinus virginianus). The remaining species not categorized as LC were mammals: eight primates (EN – Leontopithecus chrysomelas and Sapajus flavius, VU – Alouatta guariba, Alouatta palliata and Cebus capucinus, NT – Sapajus libidinosus, Sapajus nigritus, and Erythrocebus patas), four rodents (EN – Callistomys pictus, VU – Oryzomys laticeps, DD – Dasyprocta variegata and Galea musteloides), one carnivore (VU – Tremarctos ornatus), two even-toed ungulates (VU – Tayassu pecari and DD – Mazama americana), two odd-toed ungulates (EN – Tapirus bairdii and VU – Tapirus terrestris), and one cingulate (NT – Dasypus hybridus) (IUCN, 2021).

Protection techniques

In the reviewed studies, more than half (n = 66, 58%) tested or mentioned a range of diverse techniques used to protect crops from vertebrates (Table S3). The most frequently used control method was hunting (either with weapons, dogs, or traps), which was mentioned in 37 (56%) of the studies that mentioned protection techniques (Fig. 6). The second most widely represented technique was another type of lethal control, the use of poisons, which was reported in 22 (33%) studies (Fig. 6). The following poisonous substances have been reported: herbicides, rodenticides, organophosphates, sodium monofuroacetate, coumarin, pyriminil, diphacinone, biorat, estricinina, methyl bromide, metomil, aluminum phosphate, zinc phosphide, parathion, chlorpyrifos, monocrotophos, endrin, mevinphos, dicrotophos, CPT, CPTH, thallium sulfate, coumatetralyl, brodifacoum, thiodicarb, and carbofuran. Agricultural practices were mentioned in 20 (30%) studies, including field clearing, time of planting or harvest, changing the location or the type of crops, altering the density of the crops, using barrier crops or firebreaks, and providing alternative food sources. Acoustic deterrents were reported in 15 (23%) studies (Fig. 6), including firecrackers, gas cannons, firearms, yellings, sirens, predator sounds, distress calls, and horns. Visual deterrents were used in 14 (21%) studies (Fig. 6) and consisted of scarecrows, reflective objects, smoke, fire, flags, predator outlines, balloons, calcium carbonate paint, and carpenter’s chalk. Chemical repellents, including anthraquinone, methiocarb, methyl anthranilate, bidrim, thrimethacarb, dimethyl, methyl anthranilate, synergized aluminum, ammonium sulfate, copper oxalate, copper oxychloride, condensed tannins, avitrol, and soap, were reported in 12 (18%) studies (Fig. 6). Vigilance by people or guard dogs was mentioned in 12 (18%) studies. Physical barriers, such as nets, fences, electric fences, trenches, metal bands, and wire mesh, were included in eight (12%) studies (Fig. 6). Biological control in the form of introducing infectious diseases, introducing or attracting predators, or reducing suitable habitats was mentioned in six (9%) studies (Fig. 6). Reproductive control and olfactory repellants were used in four (6%) studies (Fig. 6). The types of reproductive controls mentioned were the use of sterilants, nest burning, and egg destruction. The olfactory repellents mentioned were creolin, Tabebuia extract, burnt rubber, and human odor. Capture and relocation were mentioned in two (3%) studies (Fig. 6). Finally, Capsicum was used as a palatable deterrent in one (2%) study (Fig. 6).

Figure 6 Number of studies that used or mentioned each type of crop protection technique.

Grey color on the bars represents the proportion of studies that did not determine the effectiveness of the protection techniques, magenta represents the proportion of studies that determined the protection techniques to not be effective, and green represents the proportion of studies that determined the protection techniques to be effective.

Quality of the information reported

The methodology used to identify vertebrate taxa responsible for crop-feeding and to quantify the damage to crops varied greatly among studies. The most common identification methods used were direct observation and interviews with farmers (n = 40 and 36 studies, respectively), followed by the interpretation of indirect signs (n = 24). The least used methods were trapping or hunting (n = 14), looking at stomach or crop contents (n = 8), using of previous knowledge (n = 9), expert reports (n = 7), camera traps and radiotelemetry (n = 3 each), using distribution maps and museum specimens (n = 2 each), and using stable isotope analyses, farmer complaints and hunter reports (n = 1 each). The most common method used to quantify crop damage was measuring the proportion of damaged crops (i.e., area, plants, fruits, and production), which was used in 37 studies. Interviews with local farmers were used to estimate the crop damage in 22 studies. Eight studies estimated the economic cost of crop destruction, six examined stomach or crop contents, and two studies used models to predict damage.

Of the 66 studies that mentioned the use of protection techniques, 25 indicated their effectiveness, while the other 41 studies only listed or alluded to the damage control methods used in their settings (Fig. 6, Table S3). Of the 25 studies that evaluated the effectiveness of the protection techniques, only 10 performed experiments and reported their results. The remaining studies either conveyed effectiveness by asking farmers in surveys (eight studies), or through author discussions of the effectiveness of the control methods (seven studies).

Discussion

Our literature review on crop damage by vertebrates across Latin America showed that (1) despite an increase in the number of studies published in the last decade, this research topic has been largely overlooked in the region; (2) several vertebrate taxa are involved in crop-feeding, but only a few orders are widely represented in the reviewed studies; (3) despite the wide range of crop protection techniques, lethal control by hunting or poisoning remains the most prevalent; and (4) only a fraction of the studies that mentioned protection techniques measured their effectiveness, and only a minority performed scientific experiments. We first describe the geographical and temporal distributions of the studies and then explore the type of studies, interactions between vertebrates and crops, and protection techniques. Finally, we discuss further directions and implications of management that could help reduce crop damage and human-vertebrate conflicts in Latin America.

Geographic and temporal distribution of studies

Our results showed that crop damage by vertebrates in Central and South America did not receive much attention in the published literature before 2000, with most studies being published after 2011. Considering the overall 2.3 fold increase in scientific literature worldwide from 1,067,910 articles in 2000 to 2,554,373 articles in 2018 (World Bank Data, 2021b), the number of articles on vertebrate crop damage in Latin America is increasing rapidly. Despite this increase in published articles, it is still an emerging discipline considering the projections of cropland expansion (Riahi et al., 2017), and the fact that South America has been the world’s leading region in cropland cover expansion over the past two decades (Popkin, 2022).

The country with the highest number of studies was Brazil, which is likely a consequence of its large territorial area and extensive research. Brazil is the largest country in Central and South America and the largest producer of studies in scientific and technical journals (60,148 in 2018, placing 18th worldwide) (World Bank Data, 2021b). Another important factor may be its crop production rates; data for crop production (cereals fruits, vegetables, sugar crops, roots and tubers, tree nuts, fiber crops, and oil crops) in 2019 reveals that Brazil has the highest crop production rates among all Latin American countries (FAO, 2021). Argentina and Mexico were the next two countries with the highest number of studies; both are also large countries with high scientific and crop production rates (FAO, 2021; World Bank Data, 2021b). Studies in these two countries have focused on specific vertebrate groups. All but two of the studies conducted in Argentina focused on crop damage by birds, with two species being the most frequent: the eared dove (Zenaida auriculata) and the monk parakeet (Myiopsitta monachus). Nearly half of the studies from Mexico have focused on agricultural rodent pests.

The Costa Rican case is interesting because it has a small territorial area and scientific production compared with other countries in Central and South America (only 507 articles were published in 2018) (World Bank Data, 2021b). It was also placed near the bottom of almost all crop categories evaluated in 2019 (FAO, 2021). Despite this, a relatively large number of studies have been published on crop damage by vertebrates from the country. Thus, our review indicated a higher interest in research on crop-feeding by vertebrates in Costa Rica than in other Latin American countries.

It is important to highlight that the number of published studies in a country is not directly proportional to the corresponding severity of the crop damage by vertebrates, as many smaller countries have little-to-none scientific production but have the conditions to potentially be severely affected by this type of human-wildlife conflict. Countries such as the Guianas, most Central American countries, and many Caribbean island nations have few or no published studies on the topic, but are rich in wildlife biodiversity (Mittermeier et al., 2011; Myers et al., 2000), and have high rates of poverty (Fisher & Christopher, 2007; World Bank Data, 2021a). Crop-feeding is an ecosystem disservice derived from high biodiversity (Ango et al., 2014; Naughton-Treves et al., 2003), and it can damage the economy of vulnerable communities (Gontse, Mbaiwa & Thakadu, 2018; Raphela & Pillay, 2021). The combination of these factors increases the risk of conflict between biodiversity conservation and food security (Molotoks et al., 2017). Therefore, future research on crop-feeding should focus on these countries.

Type of plantations and studies

More than half of the studies in our review were concentrated in commercial crop plantations. This could be expected because commercial plantations generally have much larger areas (Felix et al., 2014; Lima et al., 2019; Lobão & Nogueira-Filho, 2011) than smaller subsistence plantations (Can-Hernández et al., 2019; Chaves & Bicca-Marques, 2017; Naughton-Treves et al., 2003). Moreover, crop losses on subsistence plantations tend to be better tolerated by landowners, as their main objective is not linked to profit (Chaves & Bicca-Marques, 2017; Rocha & Fortes, 2015; Spagnoletti et al., 2017). Subsistence farmers also tend to be more knowledgeable about the wildlife that inhabits their fields, and they engage in practices that are beneficial to conservation (Silva-Andrade et al., 2016). However, there are also examples of communities that engage in hunting to defend their subsistence crops (Can-Hernández et al., 2019; Cossios, Ridoutt & Donoso, 2018).

Of all reviewed studies, 37% evaluated the magnitude of crop damage by vertebrates, but only 14% focused on crop protection techniques. Moreover, a large proportion of studies used interviews with local farmers to collect data and evaluate their perceptions. This method was the second most commonly used method for identifying crop-feeding vertebrates and quantifying crop damage. However, interviews were only corroborated by alternative methods in a few studies, which could present an inherent bias in the reported crop damage. Involving local communities and stakeholders in research can have positive effects on nature conservation (Beierle & Konisky, 2001; Young et al., 2013) and enable data collection over large areas (Michalski et al., 2020; Michalski & Peres, 2017). Farmers’ perceptions and knowledge are central to studies on crop damage and conservation strategies. However, relying solely on farmers’ perception of crop damage can be misleading, as their ideas of which species are responsible for damaging crops or the extent of losses may not accurately represent the reality (Albarracín & Aliaga-Rossel, 2018; Flores-Armillas et al., 2020; Hill, 2004) or be proportional to the scale of the problem (Simonsen, Tombre & Madsen, 2017). Therefore, relying almost exclusively on interviews with local farmers for data generation may result in an incorrect assessment of the conflict, which when coupled with an exaggerated perception of damages caused by vertebrates may lead to an increase in the use of lethal methods for retaliation (Can-Hernández et al., 2019). Studies that perform field validation of crop damage are important, and more effort towards using field validation methods must be made in future studies.

Vertebrates and crops

Among the 272 vertebrate taxa that were identified in the studies as causing crop damage, Rodentia was the order of vertebrates with the highest importance value. Rodents have long been considered as some of the worst crop pests worldwide (Capizzi, Bertolino & Mortelliti, 2014; Lauret et al., 2020; Stenseth et al., 2003). This concurs with our results, where they were shown to cause damage to the highest number of crop genera (45), affecting corn the most (Felix et al., 2014; Ferraz et al., 2003). Early studies on crop damage and pest control in Latin America focused on rodents (Espinoza & Rowe, 1979), and they have continued to be the main focus of research during the period included in our review (Felix et al., 2014; Ferraz et al., 2003; Sánchez-Cordero & Martínez-Meyer, 2000; Santos, 2018). Rodents can cause extensive damage to crops and have often been the target of lethal control (Hilje, 1992; Villafaña Martín et al., 1999). Four rodent species (paca – Cuniculus paca, capybara – Hydrochoerus hydrochaeris, and hispid cotton rat – Sigmodon hispidus, and the black rat – Rattus rattus) appeared in the largest number of studies.

The second order of mammals with the highest importance values was Carnivora, which interacted with 21 crop genera. Three species (Nasua narica, Nasua nasua and Procyon lotor), all belonging to the Procyonidae family, have been recorded in several studies. These species are often among the most concerning to farmers (Castillo-Chinchilla et al., 2018) and among the most damaging to crops, particularly to corn (Can-Hernández et al., 2019; Flores-Armillas et al., 2020).

The Artiodactyla order affects 18 different crop genera, but interacts mostly with corn and manioc, often causing extensive damage (Abrahams, Peres & Costa, 2018; Pérez & Pacheco, 2014; Romero-Balderas et al., 2006). Among the even-toed ungulates, two species (collared peccary – Pecari tajacu and wild boars – Sus scrofa) had the highest number of appearances in all studies. Wild boars are invasive in many parts of the world, including in Latin America, and cause extensive crop damage worldwide (Bevins et al., 2014). They can have deleterious effects on native biodiversity around the globe, even driving some species to extinction (Risch, Ringma & Price, 2021). In our review, all studies focusing on wild boars were from Brazil, where boars have been found to dominate local communities shortly after the invasion (Doutel-Ribas et al., 2019) and consume large amounts of cultivated grain (Cervo & Guadagnin, 2020). Lethal methods for wild boar control have been legal in the country since 2013, and hunting has become widespread (Rosa, Wallau & Pedrosa, 2018). Most farmers agree that this species should be eradicated (Pereira, Rosa & Zanzini, 2019).

Primates were the order that interacted with the second-highest number of crop genera (43), with corn being the top crop interaction followed by sugarcane and bananas. Primates feeding on crops were often perceived as tolerable by farmers, and they rarely used lethal control measures against them (Chaves & Bicca-Marques, 2017; Lins & Ferreira, 2019; McKinney, 2019; Rocha & Fortes, 2015; Spagnoletti et al., 2017). This might be due to them often targeting crops that are not used commercially, which could favor a peaceful coexistence between humans and non-human primate crop-feeders (Chaves & Bicca-Marques, 2017; Rocha & Fortes, 2015; Spagnoletti et al., 2017). The tolerance of crop-feeding by primates might have also been motivated by their resemblance to humans, which causes empathy (Dore, Eller & Eller, 2018; Rocha & Fortes, 2015). Lethal control of primates was only recorded in the case of invasive vervet monkeys (Chlorocebus aethiops) in Barbados, where they cause damage to a variety of crops, and campaigns to reduce their population have been conducted (Boulton, Horrocks & Baulu, 1996). Lethal control of primates is also infrequent in Africa and Asia, where most farmers use non-lethal techniques (Marchal & Hill, 2009; Mc Guinness & Taylor, 2014; Siljander et al., 2020).

Like rodents, birds have long been considered agricultural pests, and the damage they cause to crops is a global concern (Anderson et al., 2013; de Mey, Demont & Diagne, 2012; Kale et al., 2014; Montràs-Janer et al., 2019). These perceptions have often motivated lethal control methods in an effort to reduce bird populations; however, these attempts are often unsuccessful (Linz et al., 2015). Among the studies included in this review, the trend of negative perceptions by farmers and their usage of lethal or reproductive control has continued (Basili & Temple, 1999; Bucher & Ranvaud, 2006; Canavelli, Swisher & Branch, 2013), although in some cases, non-lethal protection techniques have been tested with positive results (Avery, Tillman & Laukert, 2001; Robles et al., 2003). Some studies have found that bird species that feed on crops, such as sheldgeese (Chloephaga sp.) or mourning doves (Zenaida macroura), offset their negative impact by also feeding on weeds, which benefits crop production (García & Peiró, 2016; Gorosábel et al., 2019).

Among birds, Passeriformes had the highest importance value, being the second most recorded order after Rodentia. Despite this, the only passerine species that appeared in more than four studies was the house sparrow (Passer domesticus). Passerines interacted with 26 crop genera, but most of the damage was concentrated in rice, corn, and sorghum. Columbiformes have been reported to cause frequent damage to a wide range of crop genera, including corn, sorghum, wheat, soy, rice, and sunflowers. Three of the most prominent crop pests in Latin America belong to this order: the eared dove (Zenaida auriculata), which appeared in 15 studies, and two species of pigeons (Patagioenas maculosa and P. picazuro), which cumulatively appeared in 11 studies. Damage by Psittaciformes was concentrated mostly on corn, followed by sunflower and sorghum. Another one of the main bird pest species on the continent is the psittacine monk parakeet (Myiopsitta monachus), which was reported in 11 studies. These three pests (doves, pigeons, and parakeets) cause extensive damage to agricultural crops in many countries. However, studies on them have mostly been conducted in Argentina and Uruguay, where they have been the subject of many damage control methods (Bruggers, Rodriguez & Zaccagnini, 1998; Canavelli, Aramburú & Zaccagnini, 2012). Anseriformes were mostly reported to damage wheat and rice in their wintering areas, causing conflicts with local farmers (Gorosábel et al., 2019).

The order with the most threatened species was Primates. Thus, it is a good prospect for their conservation that farmers in Latin America tend to tolerate crop-feeding by primates and seldom use lethal control against them (Chaves & Bicca-Marques, 2017; Lins & Ferreira, 2019; McKinney, 2019). Most of the species that are not considered of least concern are infrequent in the literature, with the most frequent being the Andean bear (Tremarctos ornatus) and white-lipped peccary (Tayassu pecari), which appeared in four and three studies, respectively. Although not frequently cited in the revised literature, some of these species have been reported to cause extensive damage or are of great concern to farmers. For example, in a Peruvian study, the Brazilian tapir (Tapirus terrestris) was an infrequent crop-feeder but caused the largest proportion of damage per affected field, and it was hunted by locals to offset crop losses (Naughton-Treves et al., 2003). The cacao-rat (Oryzomys laticeps) caused the most damage and generated the highest number of complaints from farmers in the Brazilian Atlantic Forest, where farmers used lethal control methods against it (Lobão & Nogueira-Filho, 2011). White-lipped peccary (Tayassu pecari) causes damage to corn plantations in the Brazilian state of Mato Grosso, and farmers periodically cull the local population using firearms, traps, and mass poisoning (Lima et al., 2019). Lastly, the Andean bear (Tremarctos ornatus) caused little damage to banana and plantain crops in Colombia but generated strong negative attitudes among locals towards their presence and conservation efforts (Escobar-Lasso et al., 2020). The human-wildlife conflicts involving these threatened species may hinder conservation efforts by reducing the tolerance of local farmers and motivating lethal control.

Protection techniques

In our review, many types of crop protection techniques have been reported, but lethal control of crop-feeding populations via hunting and poisoning was the most commonly used protection method. Farmers may turn to lethal control after trying other protection techniques without success (Lima et al., 2019), and they tend to perceive hunting or poisoning as the most effective damage control method (Abrahams, Peres & Costa, 2018; Canavelli, Swisher & Branch, 2013; Lima et al., 2019). However, very few studies have provided reliable evidence that lethal control effectively reduces crop damage. Of the 37 studies that reported hunting as a control measure, only 10 evaluated its effectiveness, and only one managed to perform experiments. Pérez & Pacheco (2014) reported a reduction in crop damage (from 27.61% to 4.59%) in hunted crop fields when compared to control plots, but the effectiveness of hunting was only slightly higher than that of non-lethal alternatives (combination of agricultural practices, olfactory and visual deterrents, and vigilance).

The use of hunting as the main technique for reducing crop damage poses a series of problems. First, many of the most prevalent crop-feeders are species that have short life cycles and high reproductive rates, such as rodents, which enables them to recover faster from reductions in population size (Hein & Jacob, 2015). Hunting may also cause targeted species to modify their movement patterns and activity regimes (Béchet et al., 2003; Keuling, Stier & Roth, 2008; Little et al., 2016; McGrath, Terhune & Martin, 2018), which may alter the area and intensity of crop damage. Finally, trapping has been reported to be the most effective way of hunting to reduce vertebrate populations, but acquiring and maintaining traps can be expensive and labor-intensive (Rosa, Wallau & Pedrosa, 2018). Even in situations where hunting is not an effective method for protecting crops, it can provide farmers with alternative sources of food or income (Naughton-Treves et al., 2003) or grant social status (Cossios, Ridoutt & Donoso, 2018), which could explain its popularity as a protection technique among farmers.

The effectiveness of the use of poisons to control crop damage was evaluated in nine of the 22 studies that mentioned it. Six of the studies deemed that the use of poisons was effective in reducing crop damage, three of which performed experiments. Generally, poisons are considered an easy way to reduce populations of crop-feeding species, but their use can cause several environmental problems, such as non-target or vulnerable species being seriously harmed. Lima et al. (2019) reported that hundreds of white-lipped peccaries (Tayassu pecari) were killed simultaneously through the use of poisonous substances. Similarly, bird species that roost in large groups, such as Dickcissels (Spiza americana), can be killed in great numbers when their nesting sites or watering holes are poisoned (Basili & Temple, 1999). Furthermore, poisons can have severe consequences for carnivores and scavengers that feed on the carcasses of poisoned animals (Baudrot et al., 2020; Kalaivanan et al., 2011). In addition to the dangers that chemical pesticides pose to the environment, they can also pose a serious threat to the health of human workers and consumers (Rani et al., 2021).

To protect the environment and native wildlife alongside the interests of local communities, alternative methods for pest control must be tested and developed. From our review, only seven studies performed experiments to test the effectiveness of non-lethal crop protection techniques. Wire mesh exclosures significantly reduced damage by wildlife to manioc and walusa but not to corn in Bolivia (Pérez & Pacheco, 2006). Some laboratory experiments on Dickcissels captured in Venezuela tested the effectiveness of chemical repellents in reducing rice consumption, and found that both methiocarb and anthraquinone reduced consumption by 70% (Avery, Tillman & Laukert, 2001). Mitchell & Bruggers (1985) tested the effectiveness of methiocarb as a chemical repellent, as well as that of an olfactory (Tabebuia extract) and visual (blue carpenter’s chalk) deterrent, in reducing damage to cacao by woodpeckers in the Dominican Republic, but their results were inconclusive. Rodriguez et al. (1995) compared the effectiveness of methiocarb with that of a visual deterrent (calcium carbonate paint) in reducing eared dove damage to sunflowers and found that the latter was much more effective. Robles et al. (2003) found that using reflective objects as visual deterrents was more effective in reducing bird damage to quinoa than the chemical repellent bidrim. The effectiveness of a palatable repellent (Capsicum) and an olfactory repellent (Creolin) in reducing wildlife damage to corn in Colombia was tested, but no significant differences between the treatments and controls were found (Castillo-López et al., 2017). Thus, we believe that the use of non-lethal control techniques has been insufficiently tested and should be explored further to benefit the protection of biodiversity and the safety of commercial and subsistence crop plantations.

The results of our literature review point to a gap in the knowledge about vertebrate-crop conflicts in Latin America. However, it is important to highlight two aspects of the methodology we used that could bias our results. First, we did not include most kinds of gray literature in our review, and there may be more knowledge on the topic to be found in reports, or MSc and Ph.D. theses that are not published in scientific journals. However, conference proceedings were included in our review to minimize this bias (McAuley et al., 2000). Another shortcoming of our methodology is that we only performed searches using terms in English. While this is the main language used for scientific communication and publication, it is not the predominant language spoken in Latin America. We included studies returned by searches using terms in English but written in Spanish, Portuguese, or French. However, there could be more studies on this topic that would only be found by performing searches using terms in the languages spoken in Latin America. Despite these limitations, we believe that our review offers an accurate depiction of the published scientific literature on crop damage by vertebrates in Latin America.

Implications for management and future directions

Human-wildlife conflict is a pressing issue due to the simultaneous reduction of natural spaces and global human population growth. Crop damage is one of the most prominent reasons for conflict, as it affects the food security and economy of local communities. Despite this, research on crop-feeding by vertebrates in Central and South America is still emerging, and the body of literature on this topic is still limited. There is a lack of standardized methodologies to perform studies on crop damage, overreliance on farmers’ perceptions, and a lack of consensus on which protection techniques are preferable. As per our review, only 10 studies in the last four decades have performed experiments to test the effectiveness of crop protection techniques, seven of which tested non-lethal methods. Farmers tend to prefer lethal control methods that can endanger vertebrate populations, harm the environment, and negatively affect human health. We consider that there is an increased need to test non-lethal crop protection techniques in Latin America, which is already happening in Africa or Asia. Reliable and extensive experimentation should be conducted in different settings across Latin America to test which techniques work on different groups of vertebrates and crops that are involved in crop-feeding in the region.

Finding techniques that effectively protect crops from vertebrates without causing mortality is essential for solving this conflict while preserving both the environment and the interests of local communities. However, crop protection alone will not be able to solve this problem, and is only treating the symptoms and not the cause of the problem. Effective non-lethal protection methods need to be combined with a reduction in natural habitat loss and fragmentation, so that wild animals do not have to turn to agricultural products for food. Pairing effective non-lethal crop protection techniques with the conservation of native vegetation will reduce human-wildlife conflicts and help improve the quality of life of local communities while protecting native wildlife. Implementing such measures may present a difficult challenge owing to the current and predicted human population growth, current systems of production, and consumer-commodity model-based economies that demand increasing amounts of land for food production worldwide. Global agricultural development requires a systemic shift towards a more sustainable model that reduces the competition between food production and wildlife for land and resources.

Conclusions

Research on crop damage by vertebrates in Latin America is scarce; however, our review of the published literature provides some relevant insights. Most of the studies published in the last four decades were concentrated in only a few countries (Brazil, Argentina, Mexico, and Costa Rica), and we suggest that studies on this subject should be carried out in other Latin American countries that could potentially be greatly affected by crop-feeding. Vertebrates from 23 orders were involved in crop-feeding, and eight of them were the most represented (Rodentia, Passeriformes, Columbiformes, Carnivora, Psittaciformes, Artiodactyla, Primates, and Anseriformes). Damage was reported to 77 genera of crops, but most interactions were concentrated on just 10, with corn being the most prominent. Lethal control methods are favored by farmers and are perceived as the most effective way to reduce vertebrate crop damage. However, most studies have not quantified the effectiveness of protection techniques, and only a minority have tested protection methods through experimentation; many of them solely rely on farmers’ perceptions. Lethal control can have negative consequences on wildlife, the environment, and human health. There is a need to find effective non-lethal protection techniques that minimize damage to wildlife and protect local economies. To achieve this, methodologies for studying crop-feeding need to be standardized, and widespread experimentation needs to be performed across Latin America and other regions across the globe.

Supplemental Information

Supplemental Information 1 List of the 113 reviewed studies with geographical data.

Including information on the type of study that they are, their country, whether coordinates and a map of the study area are provided, the coordinates of the locations plotted in Fig. 3, and notes about how these coordinates were obtained.

Click here for additional data file.

Supplemental Information 2 List of vertebrate species reported to produce crop damages across the 113 reviewed studies.

Including information on the number of studies they appear on, the crop genera they interact with and the protection techniques that have been used on them.

Click here for additional data file.

Supplemental Information 3 List of the 113 reviewed studies with species and crop protection data.

Including information on the crop taxa in each study, the vertebrate taxa that interact with them, the protection techniques used, and the efficiency of the protection techniques.

Click here for additional data file.

Supplemental Information 4 PRISMA 2020 checklist.

Click here for additional data file.

Supplemental Information 5 Raw data.

Click here for additional data file.

We are grateful to Andreas Kindel, Júlio César Bicca-Marques, Mendelson Guerreiro de Lima, Caio José Carlos and three anonymous reviewers for comments and suggestions on an early stage of this manuscript. We thank Stephen Gillanders for his help with English language editing and proofreading.

Additional Information and Declarations

Competing Interests

Author Contributions

Data Availability

Fernanda Michalski is a research associate from Pro-Carnivores Institute. The authors declare that they have no competing interests.

Adrián Alejandro Cuesta Hermira conceived and designed the experiments, performed the experiments, analyzed the data, prepared figures and/or tables, authored or reviewed drafts of the paper, and approved the final draft.

Fernanda Michalski conceived and designed the experiments, analyzed the data, prepared figures and/or tables, authored or reviewed drafts of the paper, and approved the final draft.

The following information was supplied regarding data availability:

The raw data used in all analyses is available in the Supplemental File.

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
