# Peer review of "Crop damage by vertebrates in Latin America: current knowledge and potential future management directions"

_PeerJ, doi:10.7717/peerj.13185_

## Round 0.1 · original submission · Major Revisions

Please consider the comments by the three reviewers. I agree that this review paper needs to include the most important references and as one of the reviewers indicated, some of them are missing. In addition, topics such as damage to crops should be added. Also, some of the orders of Mammals were not included thoroughly and for the paper to be complete all orders need to be included.

Reviewer 1 ·

Basic reporting

no comment

Experimental design

I have some doubts as to why authors included only the first 50 records from Google Scholar. This are probably the ones that have been more accessed, but not necessarilly the most recent.

I don´t think I understood what authors want to say in lines 220-223

Why were thesis and gray literature not included in the review?, they can also have relevant data and information.

I do not see the relevance of including if the research papers include the exact coordinate. I consider the authors go through this extensively.

Validity of the findings

The authors mention that relying on the perception of farmers on crop damage can be misleading as their ideas of which species are responsible for damaging crops. I think that the farmers´ day to day experience on examining the damage in their crops provides very important knowledge as to clearly identifying which species are related to the different damages.

Reviewer 2 ·

Basic reporting

The review of the literature was insufficient.

Experimental design

The choice of key terms to the search is biased towards some orders of Mammals. Considering that one of the keywords (Line 66) is “bird damage”, and that several of the references that you cited in the Introduction section are related to birds (such as: Lines 106-107: Canavelli, Swisher & Branch, 2013; Lines 107 and 141: Linz et al., 2015; Lines 144-145: Mitchell & Bruggers, 1985). Please explain why did you not use the term “Bird” in your search? Not including this term (and, for example, considering “Mammal”) may skew the results.

Validity of the findings

The results, discussion and conclusion are conditioned by the strong bias to Mammals (instead of including all Vertebrates).

Additional comments

General comments:

The study is attempts to address the current knowledge crop damage by vertebrates in Latin America through literature review. However, the articles search criteria are biased towards certain Orders of Mammals (please, see particular comment Lines 160-167). So, the list of studies included in the review is incomplete. The results, discussion and conclusion are conditioned by this bias.
At the end, I added a list of some relevant works that were missing from the review.

Particular comments:

- Introduction

Lines 84-90: These sentences refer to “pests” in general. These data correspond mainly to losses due to invertebrate pests, insects particularly. I suggest incorporating, below, background on the losses caused by vertebrates to crops.

Lines 105-116: An important topic that is missing from this paragraph is that lethal control can also generate contamination as a result of the use of poisons as toxic baits to kill vertebrate pest, and this may cause social problems. I suggest that you include that topic.

- Survey methodology

Lines 160-167: The choice of key terms to the search is biased towards some orders of Mammals. Considering that one of the keywords (Line 66) is “bird damage”, and that several of the references that you cited in the Introduction section are related to birds (such as: Lines 106-107: Canavelli, Swisher & Branch, 2013; Lines 107 and 141: Linz et al., 2015; Lines 144-145: Mitchell & Bruggers, 1985). Please explain why did you not use the term “Bird” in your search? Not including this term (and, for example, considering “Mammal”) may skew the results.

Line 175-176: Why is this clarified? According to Line 159 the search in dataset included Google Scholar already. Please, clarify.

Line 176: Was the search and review carried out by one of the authors (Line 158) or by both authors (Line 176)? Please, clarify.

Lines 210-225: Please, see previous comment Lines 160-167. Due to the terms included in the search, the results may be biased towards Mammals, instead of including all Vertebrates.

Lines 240-246: The information in the figure 1 is redundant with the information in the text.

- Results

Line 252 and figure 2: The search was carried out between May 2020 and May 2021 (Line 160). So, the year 2021 cannot be compared with the other years because only five months of the year was analyzed.

- Some examples of missing studies:

Category (1) Crop damage evaluation, if the damage caused to crops by vertebrates in the area was assessed (Lines 178-179).

- Canavelli et al. (2014) Multi-level analysis of bird abundance and damage to crop field. Agriculture, Ecosystems & Environment 197:128-136. DOI: 10.1016/j.agee.2014.07.024.
- Gorosábel et al. (2019) Evaluating the impacts and benefits of sheldgeese on crop yields in the Pampas region of Argentina: A contribution for mitigating the conflicts with agriculture. Agriculture, Ecosystems and Environment 279:33-42. DOI: 10.1016/j.agee.2019.04.002.

Category (5) Pest species or outbreak overview, if the article reports on general information about one or several species considered to be pests or on specific outbreaks (Lines 184-185).

- Bucher & Ranvaud (2006) Eared dove outbreaks in South America: patterns and characteristics. Acta Zoologica Sinica 52: 564-567.
- Bruggers, Rodriguez & Zaccagnini (1998) Planning for bird pest problem resolution: a case study. International Biodeterioration & Biodegradation 42:173-184.

Category (6): Crop-raider species behavior, if the study focused on the diet or other behavioral aspects of the vertebrate species (Lines 185-186).

- Álamo Iriarte, Sartor & Bernardos (2019) Agriculture in semiarid ecosystems favors the increase fossorial rodent’s activity in La Pampa, Argentina. European Journal of Wildlife Research 65:47. DOI: 10.1007/s10344-019-1281-7.
- Bucher (1982) Colonial Breeding of the Eared Dove (Zenaida auriculata) in Northeastern Brazil. Biotropica 14:255-261. DOI: 10.2307/2388083.
- Calamari et al. (2018) Variations in pest bird density in Argentinean agroecosystems in relation to land use and/or cover, vegetation productivity and climate. Wildlife Research 45. DOI: 10.1071/WR17167.
- Ranvaud et al. (2001) Diet of eared doves (Zenaida auriculata, Aves, Columbidae) in a sugar-cane colony in South-eastern Brazil. Brazilian Journal of Biology 61:651-660.

Reviewer 3 ·

Basic reporting

Thank you to the authors for all the work invested in reviewing this important topic. Crop damage by vertebrates is a vital issue for both wildlife conservation and human welfare, and so this geographically-focused review has cross-disciplinary relevance.

I believe the authors have slightly overstated the idea that crop foraging is poorly studied. There have been other recent reviews that are not referenced in the paper but would be of interest, such as:

• Torres DF et al (2018). Conflicts between humans and terrestrial vertebrates: A global review. Tropical Conservation Science 11: 1-15.

• Lindell C et al (2018). Enhancing agricultural landscapes to increase crop pest reduction by vertebrates. Agriculture, Ecosystem & Environment 257: 1-11.

• Hill CM (2018). Crop foraging, crop losses, and crop raiding. Annual Reviews of Anthropology 47: 377-394.

This review is still a useful contribution to the literature, however, especially since it focuses on a specific region for which agriculture is particularly important, and the review highlights the uneven distribution of studies across the countries of Latin America.

Experimental design

Your inclusion/exclusion criteria for the systematic review are generally well explained. Figure 1 especially is very easy to follow. Where I got a bit confused is in the choice of search terms. It says that every possible combination of the following terms (lines 161-167) were used, but surely *most* of the output of this method would be irrelevant to the study? Did you perhaps have these terms grouped into categories (so all possible combinations from each field) or were there literally searches like “Panama AND Honduras”? This section could be explained more clearly. Likewise, I am a little surprised that this many search combinations over a 40-year period yielded only 59 papers to review. What was the number of papers generated before the screening steps?

The paper could be stronger with some focus on organization; the results and discussion blend into each other which means at times the paper is unnecessarily repetitive. As a systematic review (rather than a more general lit review), the results section should be fairly distinct from the discussion. The first paragraph under “results” (lines 239-246) would probably fit better in the Methods section, and the limitations section (“Quality of information reported”, lines 341-367) would be better suited to the discussion section.

I would recommend something more like:

-Methods (search terms, inclusion/exclusion criteria, number of papers returned, filtering to final sample size)

-Results (key findings, which are currently start of discussion, then sections in order of importance on what crops, what vertebrates, mitigation strategies, location)

-Discussion (lead with most important/surprising finding, talk about limitations of the study or areas for further research, finish with recommendations)

Validity of the findings

The main findings as listed in lines 370-376 are great, but I’d like to see this used more fully for setting priorities in the discussion section. For example, it’s clear from your data that Rodentia are the most frequently problematic taxon – so what can we do with that information? How can farmers best respond to rodents, compared to some other type of animals?

The work would also benefit from more nuanced recommendations to reduce crop damage. The authors are absolutely correct in their point that crop protection addresses a symptom and not a cause of human-animal conflict, but recommending further protection of more natural spaces as a solution ignores the importance of conservation in shared spaces. Farmers and small-scale landowners may have little control over national or global conservation priorities, so what can they do to reduce crop damage on their lands? Did your review uncover any particularly successful methods, such as planting buffer crops, use of guard animals, or growing crops that are undesirable to the area’s major pest species?

Finally, you’ve highlighted the fact that most of these studies are from Brazil or Costa Rica, and so a broader geographical range is important. So, which are the priority countries for future study, and why?

Additional comments

GENERAL COMMENTS

Just a thought, and there’s no pressure at all to change it…. In the primatological community, there’s been a movement in recent years to replace the term “crop raiding” with something less value-laden like “crop feeding” or “crop foraging”. The thinking behind this is that “crop raiding” implies theft, or bad behaviour, or some sort of intent to cause problems for farmers. I understand this may be more important for primates than for other vertebrates, because primates are so often and easily anthropomorphized, but it’s something you might consider.

A combination of Supplemental tables 1 and 3 would be quite nice in the paper text instead of the supplemental files.



MINOR LINE COMMENTS

Line 51: “all vertebrate orders” – does this mean all 4 vertebrate orders listed a few lines above, all 139 vertebrate orders represented in your sample, or all vertebrate orders in the world?

Line 69: “estimate of 1,591 million hectares” = so, 1.5 billion?

Line 140: Reproductive control is not a lethal method, so this sentence could be phrased more clearly.

Line 176: Why were only the first 50 records obtained by Google Scholar searches used? This seems like a flaw in the systematic review method, but if it was part of the selection design it should be justified here.

Line 306: “Of the 114 taxa…” - please insert “vertebrate taxa” here, because it’s not immediately clear if you’re talking about crops or animals otherwise.

Lines 342-352: I’m not sure the precision of GPS coordinates or the presence/absence of a map in the paper is particularly important for this review.

Line 490: Why was importance values above 10 the cut-off for inclusion in the discussion? You explained how importance value was calculated in the methods, which is fine, but it might be worth going back to explain why this specific figure is the threshold for “important” in terms of the study’s conclusions.

Figure 4: Since you have two colours to distinguish between mammal and bird orders, this figure might look at bit better if you put them in a more logical order – either by descending importance value L-R or in alphabetical order.

Figure 6: The colours in this figure are not distinct enough if people happen to print the paper grayscale like I did. Could you increase the contrast between the 3 colours?

---

## Round 0.2 · Major Revisions

Even though the effort to report more literature on crop damage by vertebrates was made, an additional reviewer indicated that still there is a number of references that can be found by searching data bases such like Google Scholar and I coincided with this suggestion. Many invasive birds were not considered for example. In addition, I recommend that the paper be reviewed by a proficient English speaker or by a company, The paper needs to reach a professional level.

Reviewer 2 ·

Basic reporting

No comment

Experimental design

The criteria used by the authors discarding technical reports and theses (194-196) and searching only in English terms (lines 180-182) could overlook relevant information for the aim of review the current knowledge of crop damage by vertebrates in Latin America.
In addition that several researches about this topics is lacking still (please, see "Validity of the findings" section).

Validity of the findings

Lines 167-194: Studies about crop damage by vertebrates in Latin America are still lacking.
Just some examples:

When I tried carried out the search in Google Scholar using the terms “Bird + Crop damage + Uruguay”, I obtained as result a research which is not included in the review:
- Olivera et al. (2016) Repelentes de aves aplicados a la semilla de soja: compatibilidad con el inoculante y residualidad en cotiledones. Agrociencia Uruguay 20(2):51-60

When I tried carried out the search in Google Scholar using the terms “Bird + Crop damage + Argentina”:
- Codesido et al. (2015) Relationship between pest birds and landscape elements in the Pampas of central Argentina. Emu 115(1) 80-84. DOI: 10.1071/MU13110

Using the terms “Bird + Crop damage + Costa Rica”:
- Monge (2013) Lista actualizada de aves dañinas en Costa Rica UNED Research Journal 5(1): 111-120. DOI: 10.22458/urj.v5i1.197. In this article, the author carried out a collection of relevant data of several bird species damage on crops. For that, Monge (2013) used literature which in this work (Cuesta Hermira and Michalski) was not cited due to it is considered “grey literature”. I understand that it was their criteria, but if the authors do not use this literature, the aim of review the current knowledge of crop damage by vertebrates in Latin America is incomplete.

“Mammal + Crop damage + Argentina”:
- Jackson (1988) Terrestrial mammalian pest in Argentina - An overview. Proceedings of the Thirteenth Vertebrate Pest Conference 13:196-198. Crabb and Marsh (Eds).

“Vertebrate + Agriculture + Perú”:
- García Mendoza and Prieto Rosales (2019) Análisis preliminar de los daños ocasionados al maíz por vertebrados plagas en la localidad Pilcos, Colcabamba, Perú. Tayacaja 2(1):111-126. DOI: https://doi.org/10.46908/rict.v2i1.43

Additional comments

Additional comment 1: The number of lines that the authors indicate in the resubmission letter is inconsistent with the lines in the tracked changes manuscript. So that, the revision process was difficult.


Particular comments:

Lines 124-142: The bias towards Mammals is maintained. Now, in this new review, the authors included other Vertebrate groups, but the background are about Mammals still. For instance, there is many research with pest birds around the world.

Lines 143-144: In this new review, the number of studies nearly doubled... Do you think that this sentence must remain?

Lines 180-182 and 648-655: The aim of this study is to assess current knowledge of crop damage by vertebrates in Latin America, but the search was only in English. This assumption could did not consider key studies about this topic. Moreover, and considering that “This review can be used by a broad audience, from researchers to conservation practitioners, and from subsistence to commercial farmers” (Lines 160-161), so that, the search should consider Portuguese and Spanish the most common languages in the countries of the study area.

Lines 283-287: The sum of studies is 94, instead of 88.

---

## Round 0.3 · Minor Revisions

I appreciate the changes made to the last version, it improved the paper and it reads much more clearly. The figures are appropriate and well done. My only suggestion is an additional review of the English of the paper, to reach a satisfactory level for PeerJ.

---

## Round 0.4 · accepted · Accept

I appreciate that you sent your article to a specialized English company and that you provided the certificate. I read it through and although I am not an English speaker the language was improved.